# The Role of Serum Albumin and Secretory Phospholipase A2 in Sepsis

**DOI:** 10.3390/ijms25179413

**Published:** 2024-08-30

**Authors:** Francis H. C. Tsao, Zhanhai Li, Amy W. Amessoudji, Dunia Jawdat, Musharaf Sadat, Yaseen Arabi, Keith C. Meyer

**Affiliations:** 1Departments of Medicine, Division of Pulmonary and Critical Care Medicine, School of Medicine and Public Health, University of Wisconsin, Madison, WI 53792, USAkcmeyer@wisc.edu (K.C.M.); 2Department of Biostatistics and Medical Informatics, University of Wisconsin, Madison, WI 53792, USA; zhanhai@biostat.wisc.edu; 3Saudi Stem Cells Donor Registry and Cord Blood Bank, King Abdullah International Medical Research Center, College of medicine, King Saud bin Abdulaziz University for Health Sciences, Ministry of National Guard Health Affairs, Riyadh 11481, Saudi Arabia; jawdatd@ngha.med.sa; 4Intensive Care Department, King Abdulaziz Medical City, Ministry of National Guard Health Affairs, College of Medicine, King Saud Bin Abdulaziz University for Health Sciences, King Abdullah International Medical Research Center, Riyadh 11481, Saudi Arabia; sadatmu@ngha.med.sa (M.S.); arabi@ngha.med.sa (Y.A.)

**Keywords:** albumin, infection, inflammation, nutrition, phospholipids, phospholipase A2, serum, sepsis

## Abstract

Sepsis is caused by a dysregulated host response to an infection that leads to cascading cell death and eventually organ failure. In this study, the role of inflammatory response serum secretory phospholipase A2 (sPLA2) and albumin in sepsis was investigated by determining the activities of the two proteins in serial serum samples collected on different days from patients with sepsis after enrollment in the permissive underfeeding versus standard enteral feeding protocols in an intensive care unit. Serum sPLA2 and albumin showed an inverse relationship with increasing sPLA2 activity and decreasing albumin membrane-binding activity in patients with evolving complications of sepsis. The activities of sPLA2 and albumin returned to normal values more rapidly in the permissive underfeeding group than in the standard enteral feeding group. The inverse sPLA2–albumin activity relationship suggests a complex interplay between these two proteins and a regulatory mechanism underlying cell membrane phospholipid homeostasis in sepsis. The decreased albumin–membrane binding activity in patients’ serum was due to its fatty acid-binding sites occupied by pre-bound fatty acids that might alter albumin’s structure, binding capacities, and essential functions. The sPLA2–albumin dual serum assays may be useful in determining whether nutritional intervention effectively supports the more rapid recovery of appropriate immune responses in critically ill patients with sepsis.

## 1. Introduction

Sepsis is a “life-threatening state of organ dysfunction caused by a dysregulated host response to infection” that is induced by an excessive immune response and release of pro-inflammatory cytokines into circulation [1,2,3]. Cytokines trigger widespread inflammation, known as the systemic inflammatory response syndrome, which causes progressive cell arrest and death, tissue damage, and end-stage organ failure [4]. Sepsis is a leading cause of death among hospitalized patients and is responsible for the highest healthcare costs worldwide, especially when prolonged support in intensive care units (ICUs) is required [5,6]. Although bacterial infection is the most common cause of sepsis, other infections (viruses, fungi, and protozoa) also trigger septic response [7]. Infections that trigger sepsis can involve either very small anatomic sites or systemic infection. Although local infections or bacteremia do not necessarily lead to sepsis, whether a septic response will occur in individuals with such infections remains unpredictable. Children, the elderly, immunocompromised individuals, and patients on dialysis have the highest risk of developing sepsis, and once triggered, the consequences are life-threatening [8,9]. Early diagnosis and aggressive treatment can reduce the probability of patient mortality and improve mobility [10,11].

Circulating pro- and anti-inflammatory cytokines are generally upregulated when patients become septic, and some are suggested biomarkers in sepsis [2,3]. Other biomarkers, such as procalcitonin, soluble triggering receptor expressed on myeloid cell-1 (sTREM-1), C-reactive protein, and proadrenomedullin, have also been used for the diagnosis of sepsis, gauging severity, and monitoring critically ill patients [12,13,14]. Recent work has found that a combination analysis of the circulating white blood cell (WBC) count and monocyte distribution width (MDW) enhances the management of early sepsis and progression of Sepsis-2 and Sepsis-3 [15,16]. However, these biomarkers can be widely variable and present in both patients with sepsis and those without sepsis with a sub-acute clinical condition.

In addition to the immune response to infection, certain cytokines in sepsis trigger widespread apoptosis and contribute to disease development [4]. Cell death in sepsis may involve not only regulated cellular apoptosis and autophagy but also regulated and unregulated cellular necrosis [4,17]. When tissue homeostasis is achieved, properly regulated cell death is removed through phagocytosis [18]. However, when necrotic cell membranes rupture and degrade, intracellular components are liberated, complicating the clearance of debris and exacerbating systemic inflammation [19,20]. Patients with sepsis display increased blood mononuclear cell necroptosis, in which dead cell membranes are susceptible to hydrolysis by the lipolytic enzyme phospholipase A2 (PLA2), including secretory PLA2 (sPLA2) [21,22,23,24]. PLA2 enzymatically hydrolyzes the fatty acyl chain at the 2-position of membrane phospholipid (PL) with the release of fatty acid (FA) and lysoPL. Additionally, some of the FA and lysoPL may independently provide pro-inflammatory bioactive triggers that aggravate systemic inflammation in sepsis [25,26].

Secreted PLA2 proteins are the most abundant of the phospholipase A2 superfamily [27]. Multiple isoforms of sPLA2 have been found in both human and animal tissues, and the secretory PLA2-IIA (sPLA2-IIA or sPLA2) isoform is closely correlated with bacterial-induced inflammation [28]. High levels of sPLA2 are found in the blood of patients with sepsis; sPLA2 levels peak 1–2 days after ICU admission, followed by a decline to baseline on day 5 [28,29]. Similar changes in blood sPLA2 levels were also observed in rats following endotoxin exposure [30]. Although it has been suggested that sPLA2 may function as a bactericidal agent [27,31], circulating levels of sPLA2 also increase under non-infectious acute phase responses in individuals following extensive surgery [32]. The role of the inflammatory response serum sPLA2 to infection remains unclear. It is known that an overstressed inflammatory response to infection may lead to cell death and subsequent biochemical changes in the cell membranes, including the hydrolysis of membrane phospholipids catalyzed by sPLA2 to yield FA and lysoPL, which may further enhance inflammations and affect cellular processes [25,26]. Using fluorescent liposome assay methods, we demonstrated that the actions of serum sPLA2 on hydrolyzing membrane phospholipids were greatly affected by serum albumin [33]. Serum albumin could either protect membranes from the actions of sPLA2 or enhance sPLA2 catalytic reactions on membrane phospholipid hydrolysis, and the albumin–membrane interactions were facilitated by sPLA2 [33]. We also observed that in a few serum samples collected from three septic patients, all had high levels of serum sPLA2 activity with a deficiency of the membrane-binding activity of a specific fraction of albumin (SFA) [33]. We hypothesized that the inverse correlation of increasing serum sPLA2 activity and decreasing SFA activity reflects the underlying mechanisms of cell death and membrane repair that could be utilized as downstream molecular markers of sepsis [33,34,35]. We also hypothesized that a lack of SFA binding activity in the serum of patients with sepsis might be due to the saturation of prebound FA/lysoPL at the albumin–FA/lysoPL binding sites at which the bound FA/lysoPL were probably derived from the membranes of massive injured or dead cells [33]. The interactions between sPLA2 and albumin (SFA) may be important steps in regulating membrane phospholipid homeostasis, particularly under inflammatory conditions. In this study, we aimed to elucidate the roles of these two proteins in sepsis and assess their potential as biomarkers. This study was conducted by analyzing a series of larger serum samples collected on different days from patients with defined sepsis. This study provides insights into the underlying mechanisms of the proteins and their possible applications in sepsis diagnosis and monitoring.

Nutritional support is essential for critically ill patients, and caloric restriction has been used as a therapeutic attempt to improve patient outcomes for decades. However, numerous studies have not identified significant differences in patient outcomes in terms of mortality and the duration of ICU stay between enteral permissive underfeeding and standard feeding protocols [36,37,38,39]. On the other hand, the impacts of caloric restriction on systemic inflammation in the setting of sepsis and catabolism may be difficult to detect. Therefore, we examined whether the simultaneous determination of serum sPLA2 and SFA could differentiate permissive feeding versus standard feeding on systemic inflammation in critically ill patients with sepsis, as a proof of concept for understanding complex immunometabolism markers in circulation.

## 2. Results

Three fluorescent assays were performed in this study. The sPLA2 assay determined the activity of sPLA2 enzymes present in the serum. The SFA assay measured the fluorescence intensity changes due to the binding of membrane phospholipids to a specific fraction of albumin (SFA) in the serum, facilitated by sPLA2. The Albumin Fatty Acid-Binding Activity (Alb-FA-BA) assay determined the binding between albumin and FA and lysoPL released from membrane phospholipids catalyzed by sPLA2 and evaluated the presence of pre-bound FA in serum albumin. The three assays are detailed in the Section 4 below and were published previously [29,30,33,40,41].

### 2.1. Real-Time Determination of the Activities of sPLA2 and SFA in the Sera of Healthy Subjects and Patients with Sepsis

Figure 1A presents examples of real-time measurements of changes in fluorescence intensity (FI) during the assay of endogenous sPLA2 activity in the serum of a healthy subject (HS) (green) and serum from a patient treated for sepsis in the ICU (red). The increase in FI was due to the release of FA and lysoPL from fluorescently labeled substrate liposomal membranes catalyzed by sPLA2 [29]. Both serum samples were randomly selected. The serum of patients with sepsis exhibited a time-dependent increasing FI catalyzed by endogenous sPLA2, whereas the assay mixture containing serum of healthy subject generated a slight increase of FI with time, like the baseline of the assay mixture, indicating the absence of sPLA2 in the serum of healthy subject.

Figure 1B presents examples of the activities of SFA in each different assay mixture. The assay mixture that contained fluorescently labeled substrate unilamellar liposomes (UL) + sPLA2, UL + HS serum, or UL + Sepsis serum had similar mild FI changes as substrate UL background. However, after exogenous sPLA2 was added to the mixture of UL + HS serum (UL + HS + sPLA2), FI continually decreased in a time-dependent manner within 2 min (green solid circle). Contrarily, after exogenous sPLA2 was added to the mixture of UL + Sepsis serum (UL + Sepsis + sPLA2), FI recurrently increased in a time-dependent manner within 2 min (red solid triangle). These results clearly show a markedly opposite SFA reaction pattern of the serum of patients with sepsis compared with the serum from healthy subjects. The endogenous sPLA2 activity, which was determined in the serum sample of the patient with sepsis in the sPLA2 assay (Figure 1A, UL + Sepsis), was not detectable in the SFA assay (Figure 1B, UL + Sepsis). In addition, the amount of exogenous sPLA2 in the SFA assay was too low to generate any FI increase (Figure 1B, UL + sPLA2). The rise in FI values observed in the reaction mixture of (UL + Sepsis + sPLA2) shown in Figure 1B can be attributed to the hydrophobic effect of albumin present in the serum of patients with sepsis, as previously described [33] and further discussed in the Section 3 below. The negative FI values observed for the SFA activity in the serum of healthy subjects were caused by the quenching of the fluorescently labeled substrate liposomal membranes during the assay. In the SFA reaction, sPLA2 promoted the interaction and binding of albumin to the liposomal membranes. This led to albumin coating the membranes, which resulted in fluorescence quenching and negative FI values after adjusting for baseline values, as detailed in our previous studies [33]. Therefore, in healthy subjects, normal SFA activity resulted in negative values. In contrast, septic patients exhibited positive SFA values, indicating reduced or deficient SFA activity. This suggests that increased sPLA2 activity in the serum of septic patients is associated with decreased SFA activity, demonstrating an inverse relationship between the activities of these two proteins.

### 2.2. Correlation between sPLA2 Activity and SFA Activity in the Sera of Patients with Sepsis

All serum samples were assayed at the laboratory on the campus of the University of Wisconsin without knowing the identity of each sample except the sample’s numeric number. After all samples were assayed, the results of each sample were aligned with sample identity at the Intensive Care Department, College of Medicine, King Saud Bin Abdulaziz University for Health Sciences. The serum samples were from patients who had participated in the studies of a nutritional feeding program [35]. In the Permissive Feeding Group, samples were obtained from eleven patients; in the Standard Feeding Group, samples were collected from nine patients. Serial serum samples were collected from each patient starting on the first day of participation in the feeding study program, and then serial multiple serum samples were obtained on the third, fifth, seventh, and fourteenth days. In this study, we adjusted each date of serum sample collection using day one as the date of the patient’s admission to the ICU. This date selection was based on the goal of determining the activities of sPLA2 and SFA in correlation with the patient’s pathophysiological conditions starting at an early time of sepsis [33].

The initial rates of the sPLA2 activity and SFA activity in each serum sample were determined from each corresponding real-time measurement curve, as shown in Figure 1 and described in Section 4 (Materials and Methods). The average value of the sPLA2 activity in the Permissive Feeding Group was not significantly different from the average value of sPLA2 activity in the Standard Feeding Group (Figure 2, left). Similarly, there was no difference in the average values of SFA activity between the Permissive Feeding and Standard Feeding Groups (Figure 2, right). However, the plot of sPLA2 initial rate activity versus SFA initial rate activity of all serum samples shows an increase in sPLA2 activity alongside a decrease in SFA activity (Figure 3). While both activities show positive slopes, the slope for sPLA2 reflects an actual rise in activity. By contrast, the positive slope for SFA activity, despite shifting from negative to positive values, signifies a reduction in SFA activity, given that normal SFA activity is generally negative, as described above (Figure 1) and noted previously [33]. The statistical analysis of the plot between sPLA2 and SFA activities of the total samples from both the Permissive Group and Standard Group shows a significant correlation between sPLA2 activity and SFA (*p* < 0.0001, *n* = 88). Interestingly, after the Permissive Group and Standard Group were separately analyzed for the correlation between sPLA2 and SFA, the correlation between sPLA2 and SFA was more significant in the Permissive Group (*p* < 0.0001, *n* = 50) than in the Standard Group (*p* = 0.0499, *n* = 38).

### 2.3. Comparison of sPLA2 Activity and SFA Activity Returning to Normal Values between Permissive Feeding and Standard Feeding

Serum sPLA2 activity and SFA activity were plotted against the time in days of difference between the date of blood draw and the date of patient admission to the ICU, which was designated as day 1 (Figure 4, left and right, respectively). The plot of sPLA2 versus time in days shows that sPLA2 activity decreases with an increase in days of the patient’s stay in the ICU in the Permissive Feeding Group (Figure 4, left black line). Contrarily, the plot shows that sPLA2 activity slightly increased with the duration of the patient’s stay in the ICU in the Standard Feeding Group (Figure 4, left red). The slope for the Permissive Group compared with that for the Standard Group is significantly different (*p* = 0.0092), suggesting that serum sPLA2 returned to the normal baseline value faster in the Permissive Group than in the Standard Group. The plot of SFA activity versus time in days in the Permissive Group shows a downward trend from higher positive values toward negative values. This trend indicates that SFA activity increases and returns to normal levels (negative values) (Figure 4, right black line). Contrarily, a return to a normal value of SFA activity was not observed in the Standard Group staying in the ICU (Figure 4, right red). The slope is significantly different for the Permissive Group compared with the Standard Group (*p* = 0.046).

### 2.4. Determination of sPLA2 Activity and SFA Activity in the Sera of Healthy Subjects and Non-Infectious, Non-Septic Patients

The purpose of this study was to determine the correlation between sPLA2 and SFA in healthy subjects and non-infectious, non-septic patients. The activities of sPLA2 and SFA were determined in the sera collected from 10 volunteer healthy individuals (HS) and 26 patients clinically diagnosed with idiopathic pulmonary fibrosis (IPF) without any symptoms of infection. The sPLA2-SFA plot shows that the activities of sPLA2 in these serum samples were mostly less than 10 FI/min (average for HS, sPLA2 = 5.425 ± 2.286; for IPF, sPLA2 = 3.329 ± 5.439). The SFA activity in all serum samples from both healthy subjects and patients with IPF had negative values (average for HS, SFA = −8.029 ± 0.625; for IPF, SFA = −5.156 ± 1.189) (Figure 5). None of the HS or IPF serum samples had SFA activity in the positive range, unlike some of the serum samples from patients with sepsis (Figure 3).

### 2.5. Effects of Spermidine on Albumin–FA Binding Activity (Alb-FA-BA)

The Alb-FA-BA assay measured the capacity of albumin to bind FA released from the liposome membrane phospholipids catalyzed by sPLA2. In the assay, the amount of added sPLA2 was small and did not generate a significant increase in fluorescence intensity (FI) compared with the background. However, FI increased substantially in the presence of purified human FAF albumin in the assay mixture (Figure 6). Albumin bound and removed FAs liberated from membrane phospholipids catalyzed by sPLA2 and consequently further increased sPLA2 catalytic activity [33]. The presence of spermidine at a 1.4 mM concentration in the Alb-FA-BA assay greatly decreased albumin FA-binding activity by about 50% (Figure 6). Spermidine itself had no significant inhibitory effect in the assay compared with the baseline reactions.

We previously hypothesized that a positive value of SFA activity of the serum albumin of patients with sepsis was due to albumin fatty acid-binding sites occupied by pre-bound FA derived from the sPLA2-hydrolyzed membrane phospholipids of injured cells [33]. This study was designed to investigate the effects of spermidine on the binding of fatty acids to albumin using the previously described Alb-FA-BA assay [40,41] to elucidate the presence of pre-bound fatty acids in the serum albumin of patients with sepsis, in comparison with the effects on FAF-albumin and albumin in the serum of healthy subjects. We hypothesized that spermidine would inhibit the binding of fatty acids to albumin in the Alb-FA-BA assay if albumin fatty acid-binding sites were not occupied. In this study, the serum samples from septic patients were randomly selected from two different groups; one group of serum samples had negative SFA activity values (SP Serum (Negative SFA)), and the other group had positive SFA activity values (SP Serum (Positive SFA)), as shown in Figure 3 and Figure 4. All serum samples from healthy subjects (HSs) had negative SFA activity values (Figure 5). As expected, spermidine inhibited the FA binding activity of purified human FAF albumin by 50% (blue) and the activity of albumin in the sera of healthy subjects by 30–40% (green) (Figure 7A). Contrarily, spermidine had only about a 10% inhibitory effect on Alb-FA-BA in the sera of patients with sepsis with negative SFA activity (SP Serum (Negative SFA)) (gold), whereas it had no effect on Alb-FA-BA in the sera of patients with sepsis with positive SFA activity (SP Serum (Positive SFA)) (red) (Figure 7A). The ratio of the activity of each sample in the presence and absence of spermidine ((+Spd)/(–Spd)) from Figure 7A was calculated (Figure 7B). The ratio results clearly show that the inhibitory effect of spermidine on FA binding to albumin was greater in FAF albumin than in the serum albumin of healthy subjects and was also markedly higher than that in the serum albumin of patients with sepsis. Although the spermidine effect on the Alb-FA-BA activity between the SP Serum (Negative SFA) group and the SP Serum (Positive SFA) group was statistically insignificant, it is probably due to unmatched serum samples, which were randomly selected. This study also showed that serum albumin of patients with sepsis had much higher activity in the Alb-FA-BA assay compared with the healthy subject serum, suggesting alterations in the structure and binding properties of septic patients’ serum albumin.

The hypotheses, the methods used to test them, and the study’s results are summarized and outlined below.

Hypothesis:
**Hypothesis 1.** 
*An increase in serum sPLA2 activity is associated with a decrease in serum albumin–membrane binding (SFA) activity in sepsis, which may serve as a useful biomarker for assessing patient prognosis under different feeding strategies, such as permissive underfeeding versus standard feeding.*


**Hypothesis 2.** 
*The deficiency in albumin–membrane binding capacity in the serum of patients with sepsis is caused by the binding sites on albumin being occupied by prebound fatty acids.*

Methods:Hypothesis 1 was evaluated by employing fluorescent sPLA2 and SFA assays to examine 88 sequential serum samples collected on various days from 20 patients with sepsis. Each sample was identified by a number only, and the lab personnel conducting the assays were unaware of any information regarding the sample identities.Hypothesis 2 was tested by incorporating spermidine into the sPLA2-albumin assay mixtures to detect the presence of prebound fatty acids in the albumin–FA binding sites.Results:Hypothesis 1—The analysis of sPLA2 activity and albumin–membrane binding (SFA) activity in 88 serum samples revealed an inverse relationship between the activities of these two proteins. Elevated sPLA2 activity was associated with a reduction in SFA activity, resulting in a change in SFA activity from negative to positive (Figure 1, Figure 2 and Figure 3). The correlation between these protein activities and patient outcomes under permissive versus standard feeding suggests that they could serve as potential markers for evaluating the presence and severity of sepsis in patients (Figure 3 and Figure 4).Hypothesis 2—The spermidine experiments revealed that fatty acids are pre-bound to albumin in the serum of septic patients (Figure 6 and Figure 7).

## 3. Discussion

In sepsis, the inflammatory response involves various components, including secretory phospholipase A2 (sPLA2), which is found in the serum of patients with sepsis. sPLA2 enzymes play a crucial role in the inflammatory cascade by catalyzing the hydrolysis of membrane phospholipids of injured cells to release arachidonic acid and lysophospholipids, which themselves are bioactive lipid mediators. Bioactive lipid mediators have been found in the peripheral blood of patients with sepsis and correlate with sepsis onset [42,43]. These substances can further regulate the expression of certain cytokines and exacerbate inflammation [44,45,46]. In addition, if sPLA2-generated FA is not removed from membranes in an orderly manner, the retained hydrophobic FA can be toxic to the cells [40]. Serum albumin plays a critical role in binding and removing FA and lysoPL in the bloodstream, contributing significantly to the maintenance of lipid homeostasis and overall physiological balance. We previously reported that prior to the binding of FA and lysoPL generated from membranes to albumin, albumin needs to bind membranes first, and such binding is facilitated by sPLA2 on a specific fraction of albumin (SFA) to interact with the membranes [33]. This study again supports the dynamic interactions between albumin and membranes facilitated by sPLA2 that produced negative FI in the SFA assay (Figure 1B). As explained before [33], negative FI generated in the SFA assay reactions was due to the quenching of fluorescently labeled substrate liposomes (flu-PC ULs) coated with albumin molecules. Positive FI resulted from fluorescently labeled FA and lysoPL liberated from liposome membranes hydrolyzed by sPLA2 [29]. In the SFA assay, the amount of exogenous sPLA2 added to the assay mixture was too low to hydrolyze the liposomal membrane phospholipid to release FA and lysoPL. The positive values of SFA activity observed in the serum of patients with sepsis resulted from diminished SFA activity and the hydrophobic effects of misfolded albumin discussed below.

As we observed before [33], serum sPLA2 and SFA exhibited an inverse relationship in their activities in the serial serum samples collected from twenty patients with sepsis over multiple days during their ICU stay, consistent with our earlier findings [33]. Increased serum sPLA2 is probably derived from macrophages and Paneth cells of the gastrointestinal ileum upon stimulation by lipopolysaccharide (LPS) derived from bacteria [47,48,49]. Although some studies have demonstrated that serum sPLA2 released in response to bacterial infection may have bactericide properties, the in vivo function of bacterial-triggered sPLA2 in the circulating blood remains unclear [27,31,48,49]. As LPS triggers the release of certain pro-inflammatory cytokines, such as tumor necrosis factor alpha and interleukin 1, it also increases sPLA2 expression in certain types of cells [50,51,52]. Despite the complexity of the role of serum sPLA2 in sepsis, it is conceivable that high levels of sPLA2 are likely capable of hydrolyzing cell membrane phospholipids under inflammatory conditions associated with cell injury and death [53]. In cell injury or apoptosis, the inner plasma membrane’s negatively charged phosphatidylserine (PS) externalizes to the outer membrane, the primary signal in efferocytosis [54,55]. Negatively charged PS on the outer membrane also provides an environment suitable for the sPLA2 catalytic reaction to hydrolyze membrane phospholipids [29]. The inverse relationship between the activities of sPLA2 and SFA in the sera of patients with sepsis suggests a scenario in which serum albumin acts as a scavenger protein to remove FA and lysoPL derived from injured cell membranes as a repairing mechanism to maintain cellular membrane integrity and homeostasis prior to the process of efferocytosis under systemic inflammation. If an overwhelming unregulated immune response occurs, such as that in sepsis (known as cytokine storm), an increase in sPLA2 may yield large amounts of FA and lysoPL from membrane phospholipids of many injured cells, likely from necrotic cells. As a result, this may not only saturate albumin–FA binding sites (Figure 7A Group (SP Serum [Positive SFA])), but it may also release massive unregulated bioactive lipids. A recent study determined that elevated plasma sPLA2-IIA in patients with COVID-19 is associated with disease progression and severity as well as an increase in membrane phospholipid degradation [56].

The presence of pre-bound FA in albumin in the sera of patients with sepsis was indirectly determined by the Alb-FA-BA assay in the presence of spermidine (Figure 6 and Figure 7). In albumin–FA binding, strong hydrophobic interactions between the FA carbohydrate chain and the hydrophobic albumin–FA binding site are the major force holding the FA in the albumin–FA binding site pocket. However, the binding also requires electrostatic and hydrogen bonding interactions between albumin positively charged amino acids (Lys, Arg, and His) and negatively charged FA carboxyl group to bring FA to the albumin–FA binding site. The effective inhibition of purified FAF albumin–FA binding by spermidine suggests that spermidine, a polyamine in basic form in solution n at neutral pH, competes with albumin positively charged amino acids involved in FA-binding. The spermidine inhibitory effect was greater on purified FAF albumin (about 50%) than on the sera of healthy subjects (30–40%). This is probably because purified FAF albumin FA-binding sites were FA-free and thus had the highest spermidine inhibitory effect, whereas albumin in the sera of healthy subjects might have contained some pre-bound FA and thus had a lower spermidine inhibitory effect. Likewise, the total deficiency of a spermidine effect on the FA binding capacity in albumin in the sera of patients with sepsis with positive SFA activity suggests the saturation of pre-bound FA in the serum albumin of patients with sepsis. Interestingly, the serum albumin of patients with sepsis with SFA activity in negative values had about 10% spermidine inhibitory effects, suggesting that these albumins had unoccupied FA binding sites (Figure 7A). As we previously hypothesized [33], albumin with bound FA may undergo conformational change or misfolding that increases the hydrophobic effect on membranes, a fundamental role in facilitating the recognition, binding, and increase in the catalytic activity of the sPLA2 enzyme (Figure 7A) [40,41].

Nutritional support is an essential component of care for critically ill patients in ICU settings. Despite the notion that permissive feeding may be more beneficial to patients than a standard feeding protocol, indexes of measuring the benefits in improving patient outcomes (i.e., mortality and morbidity) and measurements of several serum inflammation biomarkers have failed to prove the benefit of permissive feeding [36,37,38,39]. However, we observed that the permissive group had a significantly more rapid decline in the activity of the inflammatory response serum sPLA2 as well as a significantly increased rise in the binding capacity of SFA (from positive to negative values) than the standard group during their period of stay in the ICU (Figure 4). This suggests that the systemic inflammation in patients with permissive feeding lessened significantly compared with that in patients with standard feeding. We suggest that subtle changes in systemic inflammation might have long-term benefits that may not be detected by conventional methods as patients receive care in the ICU. The dual assays of sPLA2 and SFA activities in serial serum samples collected from the same patients on different days were sensitive enough to detect subtle in vivo changes in systemic inflammation that could provide valuable insights into disease dynamics, treatment efficacy, and the progression of various conditions. The significant improvement in inflammatory indicators detected in the permissive feeding group could be due to reduced oxidative stress as compared with standard feeding. Of note, we previously observed that rats fed with grape extract containing a high content of antioxidant polyphenolics had a significantly lower LPS-induced sPLA2 activity level in the peripheral blood [30].

To determine whether the sPLA2-SFA inverse-coupling relationship could be observed in a non-infectious disease state that lacks evidence of extensive systemic inflammation or infection, we quantitated sPLA2 and SFA activities in sera from patients with IPF. The sPLA2 activities for the IPF cohort were like the baseline level of healthy subjects (Figure 5). However, most of the SFA activities in the IPF serum samples were lower than those in the healthy subjects, which may reflect some degrees of albumin structural modifications due to compartmentalized (lung) or even systemic oxidative stress in patients with IPF. We previously demonstrated that the oxidation of albumin decreased SFA activity but did not totally diminish the SFA activity [33]. Therefore, we suggest that the increase in sPLA2 accompanied by a proportional diminution of SFA activity reflects an acute response to the conditions of the “cytokine storm” in sepsis.

In conclusion, the structural and binding property differences of albumin in septic patients compared with healthy subjects underscore the complex biochemical changes that occur during sepsis. The overall sequence of events described suggests that fatty acids released by sPLA2 from injured cell membranes bind to albumin. The pre-bound fatty acids alter the binding activity and structure of albumin, potentially influencing cellular and physiological processes in which fatty acid and membrane bindings are critical. The alterations have important implications for both the pathophysiology of sepsis and its clinical management. Other than hypoalbuminemia in critical illness, albumin binding capacity depletion and structural alterations in sepsis could also have enormous effects on its vital transport functions, its ability to serve as an antioxidant, its anti-inflammatory properties, and its important role in regulating plasma oncotic pressure [57]. Future research that could help to confirm the applicability of the observations to other settings, including a broader population and different cohorts, would enhance the generalizability of the findings.

## 4. Materials and Methods

### 4.1. Materials

Dioleoyl phosphatidylcholine (DOPC), phosphatidylglycerol (PG), porcine pancreatic PLA2 (sPLA2), human fatty acid-free (FAF) albumin, and spermidine were purchased from Sigma-Aldrich, St. Louis, MO, USA. Fluorescently labeled phosphatidylcholine (bis-BODIPY^®^ C11-PC (1,2-bis-(4,4-difluoro-5,7-dimethyl-4-bora-3a,4a-diaza-s-indacene-3-undecanoyl)-sn-glycero-3-phosphocholine)) was purchased from Molecular Probes, Eugene, OR, USA. All chemicals used in this study were reagent grade. The buffer used for the sPLA2 assay and SFA assay was 0.01 M Tris-HCl containing 10 mM CaCl_2_ (pH 7.4).

### 4.2. Serum Samples

In this study, we conducted blind assays of serum samples collected from patients who had been treated for sepsis at the Intensive Care Department, College of Medicine, King Saud Bin Abdulaziz University for Health Sciences, King Abdullah International Medical Research Center, Riyadh, Kingdom of Saudi Arabia. Serial samples were collected at consistent intervals and analyzed using standardized protocols to minimize day-to-day variability to ensure comparability, as previously described [36]. These patients were also under well-controlled caloric and protein intake feedings, either with the method of permissive enteral underfeeding or with standard enteral feeding [36]. A total of eighty-eight serial serum samples were collected consecutively from 20 patients with sepsis on different days up to 20 days after enrollment in the PermiT trial (Permissive Underfeeding (11 patients) versus Target Enteral Feeding (9 patients) in Adult Critically Ill Patients) [36]. Each sample was labeled with the number only, and any information related to the sample identity was blind to the lab assay personnel at the University of Wisconsin-Madison laboratory. All samples were transported on dry ice, stored at −80 °C, and processed uniformly prior to the assay.

### 4.3. Preparation of DOPC-PG Fluorescent Liposome Substrate

The substrate of 2 mL of fluorescently labeled unilamellar liposomes (UL) was prepared by mixing 2 mg of DOPC, 2 mg of PG, and a trace amount of Bis-BODIPY C11-PC (28 µg) in chloroform, as described previously [33]. Briefly, chloroform from the lipid mixture solution was first evaporated to dryness under nitrogen, and lipid residues were suspended in 2 mL of Tris-sucrose buffer with vortex agitation several times at room temperature within a half hour. The suspension was cooled on ice followed by sonication to break up the multilayer liposomes to unilamellar liposomes, as detailed before [33]. The 2 mL sonicated fluorescently labeled unilamellar liposome solution (flu-PC UL) was divided into 0.2 mL aliquots and stored at −80 °C before assay.

### 4.4. Serum Endogenous sPLA2 Activity Assay

An aliquot of 693 µL assay buffer (0.01 M Tris-HCl + 10 mM CaCl_2_, pH 7.4) was transferred into a glass tube (13 mm × 100 mm) on ice, followed by the addition of 9 µL of flu-PC UL and 4.6 µL of serum sample to the tube. The tube was vortexed after each addition. Then, the glass tube was placed in a test tube rack on the bench at room temperature. An aliquot of 0.3 mL was immediately transferred from the tube to a well of a white polystyrene 96-well microplate (Porvair PS White, PerkinElmer, Waltham, MA, USA) in duplicate. Routinely, two serum samples, each prepared in separate tubes, were tested simultaneously. The microplate was immediately placed in a temperature-controlled (25 °C) fluorescence microplate reader (Synergy HT, BioTek, Winooski, VT, USA). The fluorescence intensity (FI) in each well was recorded every 10 s per cycle for 120 cycles at 485 nm excitation and 528 nm emission. An initial reading was recorded as zero time, and the activity was expressed as FI vs. time (min) after the initial reading was subtracted from each subsequent reading (∆FI). The initial rate of the reaction (FI/min) was determined from the reaction curve fitted with a second-order polynomial equation and the first-degree coefficient. The sPLA2 activity was calculated and expressed as FI/min per μL of serum.

### 4.5. Serum Specific Fraction of Albumin (SFA) Assay

#### 4.5.1. Preparation of sPLA2 Working Solution for the SFA Assay

An aliquot of 1 µL of sPLA2 stock solution (2.5 µg) (Sigma porcine pancreatic PLA2 stored at 4 °C) was diluted to 500 µL in 0.01 M Tris-HCl (pH 7.4) buffer (no calcium) (marked as sPLA2 first dilute working solution, 5 ng sPLA2/1 µL). Then, an aliquot of 50 µL of the sPLA2 first dilute working solution was further diluted to 200 µL Tris-HCl (marked as sPLA2 second dilute working solution, 1.25 ng sPLA2/1 µL). Both sPLA2 dilute solutions were prepared on the day of assay and kept on ice.

#### 4.5.2. SFA Assay

The SFA assay was based on the procedure described previously to determine the albumin–membrane binding activity facilitated by sPLA2 [33]. Prior to conducting the assay, the assay buffer (0.01 M Tris-HCl + 10 mM CaCl_2_, pH 7.4) was equilibrated at room temperature (22–24 °C), and the white polystyrene 96-well microplate was placed in the plate holder of the fluorescence microplate reader. In a blank control assay, an aliquot of 296 µL of assay buffer was added to a well of a 96-well microplate at room temperature, followed by the addition of 3 µL of flu-PC UL with agitation using a 100 µL pipette and then the addition of 1 µL of the sPLA2 second dilute working solution (1 ng). After a quick and brief agitation, the fluorescence intensity in the well was immediately recorded every 3 s per cycle for 50 cycles at 485 nm excitation and 528 nm emission. For assaying a serum sample, an aliquot of 1 µL of serum was added to a well containing 295 µL of assay buffer and stirred. The rest of the procedures were the same as those described for the control blank assay. In this assay, the sPLA2 in the second dilute working solution added to the assay mixture was too low to generate significant fluorescent intensity (FI) change. Any fluorescent intensity change observed in the assay represents the activity of albumin binding/interaction with liposome membranes mediated by the sPLA2 in the assay mixture [33]. The rate of SFA activity was calculated from the reaction curve fitted with a second-order polynomial equation and the first-degree coefficient and expressed as FI/min per μg of albumin. The albumin contents in the serum samples were determined as described previously [40].

### 4.6. Determination of Spermidine Effect on Albumin–FA Binding Activity (Alb-FA-BA)

This experiment was designed to determine the presence of pre-bound FA in serum albumin. We hypothesized that spermidine (Spd), a polyamine, can decrease albumin–FA-binding activity if the albumin–FA binding site is unoccupied but cannot decrease albumin activity if the binding site is occupied by pre-bound FA. The assay was like that described previously for the determination of albumin–FA binding activity in serum [40,41]. Prior to starting the experiment, a work solution of sPLA2 was prepared by diluting 1 µL of sPLA2 stock solution (2.5 µg) (Sigma) with 0.25 mL of 0.01 M Tris-HCl, pH 7.4 (12.5 ng sPLA2/1 µL), on ice, as described above. In the routine preparation of the experiment, an aliquot of 5 µL of serum was mixed with 15 µL of deionized water in a 250 µL conical vial at room temperature (−Spd). Similarly, in a second vial, an aliquot of 5 µL of serum was mixed with 13 µL of deionized water and 2 µL of 1 M spermidine in water (+Spd), and a third vial contained 20 µL of water only. The vials were incubated at 37 °C for 10 min and then placed on ice. An aliquot of 10 µL of each vial solution was added to a glass tube (1.3 cm × 10 cm) on ice that contained 640 µL of assay buffer (0.01 M Tris-HCl and 10 mM CaCl_2_, pH 7.4) and 10 µL of fluorescent PC liposome (DOPC-PG). Routinely, not more than four tubes were used for each assay preparation (including the assay control). After each tube was prepared, an aliquot of 30 µL of sPLA2 work solution was diluted to 600 µL of assay buffer in a reservoir on ice. An aliquot of 40 µL of diluted sPLA2 solution was transferred to each tube using a multi-tip pipette. Following agitation of each tube, the four tubes were immediately placed on a rack at room temperature. An aliquot of 300 µL was transferred from each tube to a well of a white polystyrene 96-well microplate in duplicate. In the end, each well assay mixture contained 3 μL of flu-PC UL, 1 μL of serum, and 10 ng of sPLA2 in the absence or presence of 1.4 mM spermidine. The microplate was immediately placed in a temperature-controlled (25 °C) fluorescence microplate reader (Synergy HT, BioTek). The fluorescence intensity (FI) in each well was recorded every 10 s per cycle for 120 cycles at 485 nm excitation and 528 nm emission, and the initial rate (FI/min per µg albumin) was calculated as described above. Similar experiments were conducted for fatty acid-free albumin (FAF-Alb) (80 μg/assay) to replace 1 μL of serum/assay in the absence and presence of 1.4 mM spermidine (−Spd and +Spd).

### 4.7. Statistical Analysis

The Pearson’s correlation coefficients between sPLA2 and SFA in the total group, permissive group, and standard group were calculated. The means of sPLA2 between the permissive group and the standard group were compared with a mixed effects model, adjusting for repeated measures on the same subjects. We conducted a similar analysis for SFA. The slopes of sPLA2 per day between the permissive group and the standard group were compared with a mixed effects model, adjusting for repeated measures on the same subjects. We also performed a similar analysis for SFA. The ratios (+Spd/−Spd) between FAF-Alb and healthy subject (HS) serum and between HS Serum and septic patient (SP) serum were compared using Student’s *t*-test. A *p*-value < 0.05 was considered statistically significant. All statistical analyses were performed using statistical analysis software (SAS, version 9.4, Cary, NC, USA).

## 5. Conclusions

The study highlighted the interplay between inflammatory response serum sPLA2 activity and albumin’s fatty acid (FA) and lipid binding activity in the context of sepsis. The results showed that increased serum sPLA2 activity was correlated with decreased albumin FA/lipid binding capacity during sepsis. This inverse relationship suggests a crucial regulatory mechanism affecting cell membrane phospholipid homeostasis and albumin structural alterations in systemic inflammation in sepsis. A reduction in albumin’s binding capacity and alterations in its structure during sepsis could significantly impact its essential physiological functions, potentially affecting patient outcomes. The dual serum assays for sPLA2 and albumin were capable of assessing and evaluating the benefits of permissive feeding compared with standard feeding in patients with sepsis. Overall, the findings underscore the intricate relationship between inflammatory markers like sPLA2, albumin structure and function, and nutritional strategies in the management of sepsis, highlighting the potential benefits of tailored feeding approaches.

## Figures and Tables

**Figure 1 ijms-25-09413-f001:**
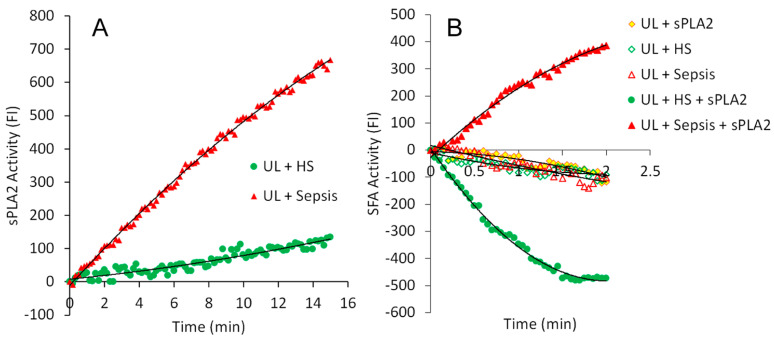
Examples of real-time measurement of sPLA2 activity (**A**) and SFA activity (**B**) in the serum. Figure (**A**) represents examples of the activities of endogenous sPLA2 in the serum of a healthy subject (HS) (green) and the serum of a patient treated for sepsis in the ICU (red). The assay reaction mixture contained fluorescently labeled substrate UL (flu-PC UL) and specified serum in assay buffer. (**B**) Represents examples of the activities of SFA in several different assay mixtures containing the components shown in the insert. In Figure (**A**), data points were collected at 10-second intervals, whereas in Figure (**B**), they were collected at 3-second intervals.

**Figure 2 ijms-25-09413-f002:**
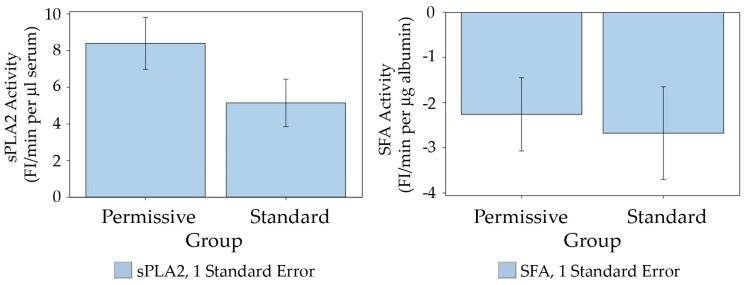
Comparison of total average activities of sPLA2 and SFA between the Permissive Feeding Group (11 patients) and the Standard Feeding Group (9 patients). Left: From Student’s *t*-test, for sPLA2, the standard group has a mean ± standard error (SE) of 5.1463 ± 1.2967 (*n* = 38). The permissive group has a mean ± SE of 8.3945 ± 1.4180 (*n* = 50). There is a marginally significant difference, with *p* = 0.1048. Right: From Student’s *t*-test, for SFA, the standard group has a mean ± SE of −2.6762 ± 1.0263 (*n* = 32). The permissive group has a mean ± SE of −2.2604 ± 0.8114 (*n* = 50). There is no significant difference with *p* = 0.7508.

**Figure 3 ijms-25-09413-f003:**
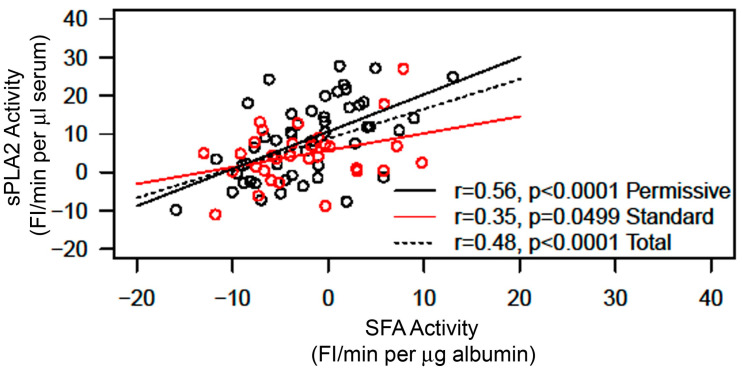
Correlation between sPLA2 and SFA of all data of both the Permissive Feeding Group (11 patients) and the Standard Feeding Group (9 patients). The Pearson correlation coefficient between sPLA2 and SFA in the whole group is 0.48 (*p* < 0.0001); in the Permissive Feeding Group, it is 0.56 (*p* < 0.0001), and in the Standard Feeding Group, it is 0.35 (*p* = 0.0499). By Fisher z transformation and standard normal test statistic, the correlation coefficient for the permissive group is significantly higher than that of the standard group, with a one-sided *p*-value of 0.0336.

**Figure 4 ijms-25-09413-f004:**
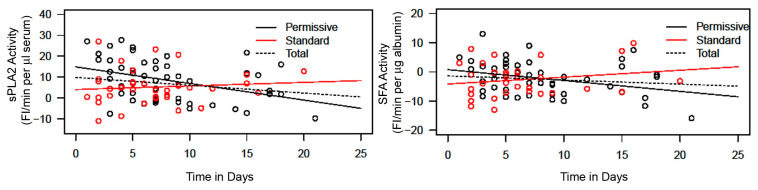
Comparison of sPLA2 activity per day (**Left**) and SFA activity per day (**Right**) between the Permissive Feeding Group and the Standard Feeding Group. Time in days represents the days between the blood draw and ICU admission (admission date as day 1). **Left**: From the mixed effects model, for sPLA2, the standard group has an intercept of 3.9902 and a slope per day of 0.1716 ± 0.3287. The permissive group has an intercept of 14.8496 and a slope per day of −0.7950 ± 0.2540. The slope for the permissive group compared with that of the standard group is significantly different (*p* = 0.009). **Right**: From the mixed effects model, for SFA, the standard group has an intercept of −4.1730 and a slope per day of 0.2360 ± 0.2162. The permissive group has an intercept of 0.7625 and a slope per day of −0.3722 ± 0.1618. The slope is significantly different for the permissive group compared with the standard group (*p* = 0.046). The line of best fit was calculated from the mixed effects model to give the overall trends of groups over days adjusting for repeated measures on the same patients.

**Figure 5 ijms-25-09413-f005:**
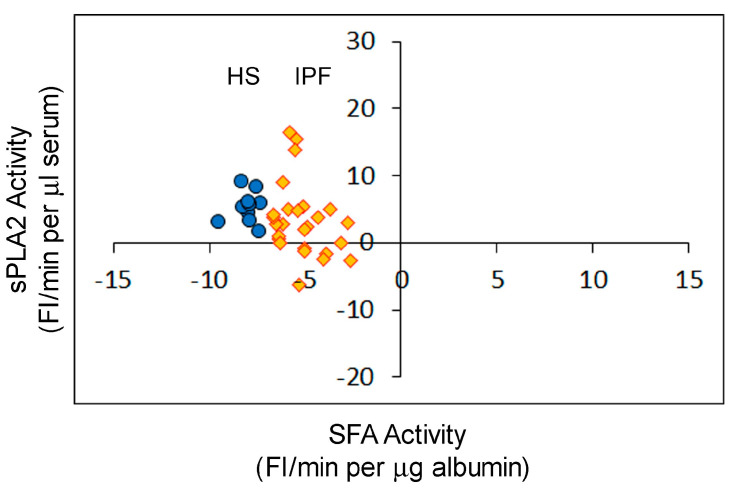
Determination of sPLA2 activity and SFA activity in the sera of healthy subjects (HSs) (blue solid circles, *n* = 10) and patients with idiopathic pulmonary fibrosis (IPF) (gold diamonds, *n* = 26). Correlation coefficient of sPLA2 and SFA in HS is 0.1155, *p* = 0.7507. Correlation coefficient of sPLA2 and SFA in ILD is −0.28438, *p* = 0.1591.

**Figure 6 ijms-25-09413-f006:**
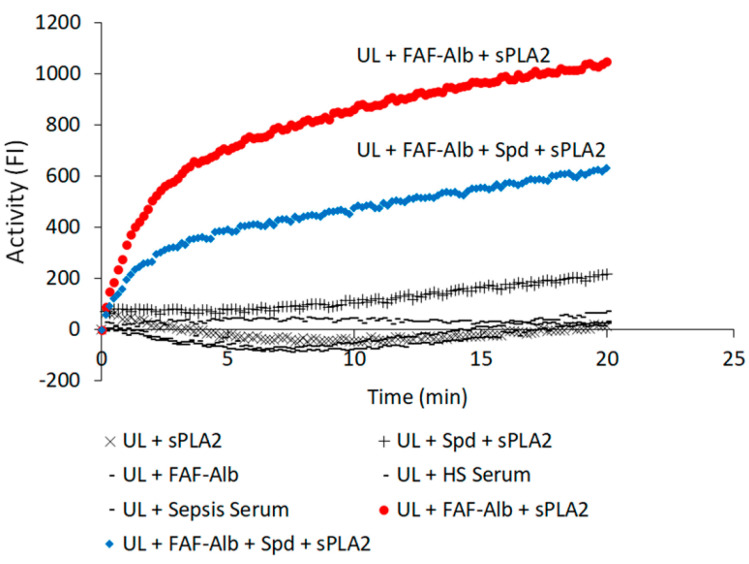
Effect of spermidine on albumin–FA binding activity (Alb-FA-BA) of purified human fatty acid-free (FAF) albumin determined by the Alb-FA-BA assay. The Alb-FA-BA of albumin was initiated by sPLA2 added to the assay mixture, which caused an immediate increase in FI in real-time measurement. Only the assay mixture that contained (substrate flu-PC UL + FAF-Alb + sPLA2) or (UL + FAF-Alb + Spd + sPLA2) yielded a significant increase in FI. The presence of spermidine in the assay mixture markedly inhibited the Alb-FA-BA of FAF-albumin. Assay mixtures that contained (UL + sPLA2), (UL + FAF-Alb), (UL + Sepsis Serum), or (UL + HS Serum) had no significant changes in FI. Abbreviations: UL, fluorescently labeled unilamellar liposomes; FAF-Alb, fatty acid-free albumin; FI, fluorescent intensity; HS, healthy subject; Spd, spermidine.

**Figure 7 ijms-25-09413-f007:**
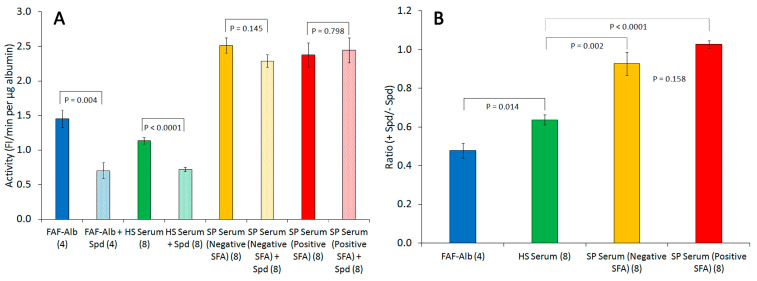
Comparison of spermidine (Spd) effects on the Alb-FA-BA of FAF albumin in the sera of healthy subjects (HSs) and patients with sepsis (SPs). (**A**) Each sample was assayed in the absence and presence of spermidine simultaneously. Each sample’s activity (FI/min per µg of albumin) was determined from its initial rate of the reaction curve, as shown in Figure 6. (**B**) The ratio of each sample activity with spermidine (+Spd)/activity without spermidine (−Spd) was calculated. The results are presented as the mean ± standard error. Spermidine inhibited purified FAF-Alb activity by nearly 50% because FAF albumin–FA binding sites were not occupied. Spermidine inhibited healthy subject serum samples by approx. 30–40%. Spermidine had a much lower or no inhibitory effect on the albumin–FA binding activity in the sera of patients with sepsis.

## Data Availability

The original contributions presented in the study are included in the article, further inquiries can be directed to the corresponding author.

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
