# Peer review of "The Role of Serum Albumin and Secretory Phospholipase A2 in Sepsis"

_ijms, 2024, doi:10.3390/ijms25179413_

Round 1
Reviewer 1 Report
Comments and Suggestions for Authors
In this manuscript, the authors describe the effects of sPLA2 and albumin in sepsis from collected serum samples in patients. The premise of the research is undoubtedly important. However, I find several critical flaws with the manuscript that must be addressed:
1. It is not clear how the serial serum samples from patients were collected and analyzed. Also, it is unclear if there is any inherent effect on the variability of collecting samples from patients from different days. Does this affect the outcome? What would be an alternative?
2. It seems surprising that the spread of data points in Figure 1 is so tight. Is there any parameter that is kept constant among patients? This needs to be made clear.
3. The p-value presented in Figure 3 would be dependent on the number of samples. If the number of data points is increased, at a certain point, everything between two distributions becomes significant. So, how is the degree of significance comparing two distributions are assessed? What other alternative statistical tests could be performed in parallel to boost this evaluation?
4. Standard error of mean or standard deviation needs to be added to the data points shown in Figure 3 and Figure 4. How is the line of best fit calculated if the patient symptoms change on different days in which samples are collected?
5. It in unclear how Figure 5 shows sPLA2 and SFA activity in healthy and IPF patients. How is activity defined and calculated in this case? I would imagine a correlation calculation would be necessary.
6. What would be the expected time at which monitored activity would plateau in each case as shown in Figure 6?
7. A general concern is that how generalizable the observations presented in this work really is. The samples are collected from a specific cohort of patients from a specific country. A discussion and/or alternative scheme to ascertain the the generalizability of these observations would be useful.
Reviewer 2 Report
Comments and Suggestions for Authors
Greeting to the authors, I have read your paper and your paper and my comments are as follows.
Firstly, the authors base this manuscript on previous research as well, however this particular manuscript from my point of view partially lacks coherence. The authors should make it clear why they chose to analyze both serum albumin and PLA2 in the same manuscript, along with practical and specific outcomes of this research. Another potential issue with this paper is that the scope of the research is not extremely specific, while the authors analyze specific targets in sepsis they should make it clear why they have chosen these markers in particular, not only because they have done so in the past.
The authors also mention "enrollment in the permissive underfeeding versus standard enteral feeding protocols in an intensive care unit." Is this a central point of the manuscript ? because the authors mention it but it hardly seems to be a focus point of the study.
The goals of the manuscript and the specific targets of this research should be pointed out, not only referring to the analysis of inflammation and sepsis in general.
The authors should make the potential applications of this research clear and concise, as an individual study and not based on previous research. As at this point the manuscript seems more like an amalgamation of previous research.
Comments on the Quality of English LanguageEnglish is fine overall.
Reviewer 3 Report
Comments and Suggestions for Authors
It is difficult to understand the aim of this study as well as its results. A clear hypothesis should be formulated: "We hypothesized that...". I suggest to the authors to make some diagram summarizing the aim/working hypothesis and the results of this study and, most importantly, the way the results of the study confirm (or not) the working hypothesis. The diagram should have a clear and logical flow.
In the last paragraph of the Results section, which is probably intended to be a conclusion, the authors assert that "The dual serum assays for sPLA2 308 and SFA evaluated the benefits of permissive feeding compared to standard feeding in 309 sepsis patients, and to show an inverse relationship with increasing sPLA2 activity and 310 decreasing albumin membrane-binding activity in patients with sepsis." - an inverse relationship of whom? How can a parameter be simultaneously in an inverse relationship with both an increasing variable (sPLA2 activity ) and a decreasing variable (albumin membrane-binding activity).
What does UW mean in "Sepsis sample days defined by UW"
In figure 2 data are characterized by mean +/- SE; however, the graphical representation is box and whiskers. There is also a p-value there - nothing is said in the legend of the figure about the statistical test used to calculate this p-value, nor is the number of items (patients) in each of the compared samples (permissive and standard) stated.
Pearson’s correlation and Student’s t-test were used - however, nothing is said about defining data distribution (normal or otherwise).
Some examples of bad English:
"appropriate programmed cell death is 66 cleared via phagocytosis"
"certain cytokines in sepsis induce 63 extensive apoptosis and pathogenesis"
"and to show an inverse relationship"
Round 2
Reviewer 1 Report
Comments and Suggestions for Authors
The revised manuscript clarifies several key critical concerns raised by this reviewer. As such, it is suitable for publication in IJMS.
Reviewer 2 Report
Comments and Suggestions for Authors
The authors have responded to my comments accordingly.
Comments on the Quality of English LanguageEnglish is fine overall.
Reviewer 3 Report
Comments and Suggestions for Authors
In figure 3 the correlations between sPLA2 and SFA is evidently positive (the slopes are positive, the correlation coefficients are positive). However the authors declare an inverse relationship between sPLA2 activity and albumin-membrane binding 353 (SFA) activity: "Analysis of sPLA2 activity and albumin-membrane binding 353 (SFA) activity in 88 serum samples indicates an inverse relationship between 354 these two activities." I am confused. Maybe I am missing something.
Before the Discussion section at the end of the Results section, the authors have introduced a summary of the hypotheses, methods, and results - please insert in the Results section of this summary references to the appropriate figures.
In the Statistical analysis section we are told that Pearson’s correlation coefficients were calculated and that Student’s t-test was employed. Nothing is said, however, about assessing data distribution. The statistical tests the authors have employed can only be applied to normally distributed data. Therefore, I ask the authors either to use non-parametric tests or to provide raw data (as Excel spreadsheets) for me to test whether the data follow a gaussian distribution - I mean the data underlying figures 2, 3, 4, 5, 7. The Excel spreadsheets should be given the titles "Figure_2", "Figure_3", ..., "Figure_7" and should be put as separate worksheets in an Excel workbook.
Round 3
Reviewer 3 Report
Comments and Suggestions for Authors
For reasons I do not understand, the authors seem to avoid directly answering my concerns. Therefore, I ask them to qualify as "correct" or "incorrect" each of the following statements. The qualifier "incorrect" should be followed by an explanation: "incorrect because...". I beg the authors not to beat around the bush, but to answer straightforwardly
First, I am referring to Figure 3. Correlation between sPLA2 and SFA of all data of both Permissive Feeding 208 Group (11 patients) and Standard Feeding Group (9 patients).
1. The slope for the Permissive group (continuous black line) is positive
"correct" or "incorrect" (if "incorrect", why?)
2. A positive slope means that the relationship between the activities of the two proteins (sPLA2 and SFA) is NOT inverse, but positive/ direct.
"correct" or "incorrect" (if "incorrect", why?)
Next, I am referring to Figure 4. Comparison of sPLA2 activity per Day (Left) and SFA activity per Day (Right) between Permissive Feeding Group and Standard Feeding Group.
3. The activity of sPLA2 (left panel) decreases with time in the permissive group
"correct" or "incorrect" (if "incorrect", why?)
4. The activity of SFA (right panel) decreases with time in the permissive group
"correct" or "incorrect" (if "incorrect", why?)
5. As the activities of both sPLA2 and SFA are decreasing with time in the permissive group, there is a positive/ direct (and NOT inverse) relationship between the activities of sPLA2 and SFA
"correct" or "incorrect" (if "incorrect", why?)
